# Effect of Xpert MTB/RIF testing introduction and favorable outcome predictors for tuberculosis treatment among HIV infected adults in rural southern Mozambique. A retrospective cohort study

Edy Nacarapa[1,2], Evans Muchiri[3], Troy D. Moon[4,5], Salome Charalambous[3,6‡], Maria E. Verdu[1‡], Jose M. Ramos[7‡], Emilio J. Valverde[8,9] *

1 Chókwè Carmelo Hospital-Daughters of Charity, Saint Vincent of Paul, Chókwè, Gaza Province, Mozambique, 2 Tinpswalo Association, Research Unit, Vincentian Association to Fight AIDS and TB, Chókwè, Gaza Province, Mozambique, 3 The Aurum Institute, Parktown, South Africa, 4 Department of Pediatrics, Division of Pediatric Infectious Diseases, Vanderbilt University Medical Center, Nashville, Tennessee, United States of America, 5 Vanderbilt Institute for Global Health, Vanderbilt University Medical Center, Nashville, Tennessee, United States of America, 6 School of Public Health, University of the Witwatersrand, Johannesburg, South Africa, 7 Department of Internal Medicine. University General Hospital of Alicante and Miguel Hernandez University of Elche, Alicante, Spain, 8 The Aurum Institute, Maputo, Mozambique, 9 Department of Medicine, Vanderbilt University Medical Center, Nashville, Tennessee, United States of America

☯ These authors contributed equally to this work.
‡ These authors also contributed equally to this work.
* EValverde@auruminstitute.org

## Abstract

### Background

Global roll out of Xpert MTB/RIF technology has resulted in dramatic changes in TB diagnosis. However, benefits in resource-limited, high-burden TB/HIV settings, remain to be verified. In this paper we describe the characteristics of a large cohort of TB patients in a rural hospital in Southern Mozambique before and after Xpert MTB/RIF introduction, together with some determinants of favorable treatment outcome.

### Methods

We conducted a retrospective cohort study of TB infected patients ≥15 years of age, diagnosed and treated at Carmelo Hospital of Chókwè between January 1, 2006 and December 31, 2017. Patient demographic and clinical characteristics, and treatment outcomes were recorded and compared before and after Xpert MTB/RIF, which was introduced in the second semester of 2012.

### Results

9,655 patients were analyzed, with 44.1% females. HIV testing was conducted in 99.9% of patients, with 82.8% having TB/HIV co-infection. 73.2% of patients had a favorable

**Data Availability Statement:** All relevant data are within the paper and its Supporting Information files.

**Funding:** The authors received no specific funding for this work.

**Competing interests:** The authors have declared that no competing interests exist.

treatment outcome. No increase was observed in the number of TB patients identified after introduction of Xpert MTB/RIF testing.

## Conclusion

Upon introduction, Xpert testing seemed to have a punctual beneficial effect on TB treatment outcomes, however this effect apparently disappeared shortly afterwards. Challenges remain for integration of TB and HIV care, as worse outcomes are reported for those patients diagnosed with TB shortly after starting ART, and also for those never starting ART. The need of reasonably excluding TB disease before ART start should be highlighted to every health care provider engaged in HIV care.

## Introduction

Tuberculosis (TB) remains a significant public health problem and the leading cause of death from infectious disease worldwide.[1,2] Autopsy studies have shown that undiagnosed TB is a common cause of mortality among HIV-infected patients.[3–6] Early detection of infected persons is fundamental to the timely provision of effective therapies, reducing transmission and TB burden.[7,8]

A rising national HIV prevalence from 2% in 1990 to 13.2% in 2015 has fueled the TB epidemic in Mozambique.[9] Mozambique is one of the World Health Organization´s (WHO) 30 high TB burden countries and faces many challenges in its successful control, especially among persons living with HIV (PLHIV).[10,11] In 2017, WHO estimated Mozambique's TB prevalence to be 551/100,000 people. However, only 52% of estimated cases were diagnosed and reported to the National TB Program (NTP), which is the agency of the Ministry of Health responsible for combating TB in the country.[12] It is estimated that nearly 60% of TB patients in Mozambique are co-infected with HIV and that TB is the most common opportunistic infection in PLHIV.[12]

Notwithstanding the high prevalence of TB/HIV co-infections in Mozambique, there are few reports about its morbidity and mortality.[13,14] Despite the evidence of the benefits of early antiretroviral therapy (ART), initiation during TB treatment[15] and the need for regular TB screening for HIV patients[16], limited information exists on the impact of undiagnosed TB at ART initiation and the influence of GeneXpert testing introduction on TB treatment outcomes.

In the present study, we aim to contribute to the body of knowledge about TB in Mozambique by describing socio-demographic and clinical characteristics of TB patients at the Carmelo Hospital of Chókwè (CHC) over a twelve-year period, assessing determinants of favorable treatment outcome, and the possible effects of introduction of Xpert MTB/RIF testing for TB diagnosis at the end of 2012. Findings from this study may support future public health interventions to address the high burden of TB in Mozambique.

## Study population and methods

### Setting

CHC is a reference hospital in the district of Chókwè, a large rural district in Gaza Province in southern Mozambique. CHC is a government hospital administered by the Daughters of Charity, Saint Vincent de Paul, Catholic missionaries since 1993. Chókwè is predominantly rural,

with a surrounding catchment population of approximately 200,000 inhabitants. Agriculture is the sole source of income for roughly 80% of the population and approximately 40% of the population migrates seasonally to South Africa, seeking work in the mines. Per the last national HIV prevalence study in 2015, Gaza Province had the highest provincial HIV prevalence in the country at 24.4%.[9]

## Study design and participants

We conducted a retrospective cohort study of TB patients ≥15 years of age, consecutively diagnosed and treated at CHC between January 1, 2006 and December 31, 2017. Per WHO guidelines, patients diagnosed at other clinics and referred to CHC for treatment and follow-up were not included in the cohort.[17]

## Data collection

Data on age, sex, residence, date of TB diagnosis, dates of anti-TB treatment (ATT) start and end, previous TB treatment, clinical presentation, HIV status at time of TB diagnosis, ATT regime, final treatment outcome, and ART status including start date and regimen were collected. Results of sputum acid fast bacilli (AFB) smear at treatment start were also recorded. When available, data on chest X-ray or Xpert-aided diagnostics were recorded.

## Anti-TB treatment regimens

In Mozambique, ATT is standard and provided free of charge exclusively by the NTP. For new patients, treatment consists of a 2-month intensive phase with a daily combination of isoniazid (H), rifampicin (R), pyrazinamide (Z), and ethambutol (E), followed by a 4-month maintenance phase with H and R. Following WHO recommendations in 2017,[18] the category II regimen for retreatment is no longer used (HRZE plus streptomycin (S) for 2 months followed by 1 month of HRZE and 5 months of HRE), and it is recommended that drug sensitivity testing is conducted in patients requiring retreatment.

## Definitions

Any diagnosis of new or previously treated active TB enrolled at CHC was considered a unique TB episode. Utilizing WHO definitions,[17] we grouped TB treatment outcomes into favorable and unfavorable categories. Documented cure or completion of ATT were considered *favorable TB treatment outcomes*. Death, treatment failure, loss-to-follow-up (LTFU), or unknown outcome were considered *unfavorable TB treatment outcomes*.

Patients with pulmonary TB (PTB) were considered to be *bacteriologically confirmed* if AFB was seen on sputum smear or if they had a positive Xpert MTB/RIF result. PTB was considered as *clinically diagnosed* if the patient had symptoms and/or findings on chest x-ray congruent with PTB, yet no AFB seen on sputum smear or a negative Xpert MTB/RIF. Extrapulmonary TB (EPTB) was diagnosed based on clinical findings and suspicion, although some cases were bacteriologically confirmed on biopsy specimens either by smear microscopy or Xpert MTB/ RIF. Patients diagnosed with both PTB as well as EPTB simultaneously were classified as having PTB.

We divided TB patients into categories according to their HIV and ART status: 1) HIV-negative; 2) HIV-positive never started on ART; 3) HIV-positive started on ART after ATT, 4) HIV-positive on ART for <90 days at ATT initiation; and 5) HIV-positive on ART for ≥90 days before ATT initiation. Rationale for the last two categories was to differentiate between 1) those starting ART with active TB missed by the treating clinician and in which a TB diagnosis

was given following the development of symptoms of immune reconstitution inflammatory syndrome (IRIS) (ATT initiation <90 days after ART initiation); and 2) those patients acquiring and developing TB when on ART (ATT initiation ≥90 after ART initiation).

## Statistical analysis

Data analysis was conducted using STATA version 14 (StataCorp, College Station, TX, USA). Descriptive statistics summarized socio-demographic and clinical characteristics. For continuous variables, median and interquartile ranges (IQR) were reported while frequency and percentages were reported for categorical variables. Bivariate associations between treatment outcomes and categories of socio-demographic and clinical characteristics were analyzed using chi-square test of association. Multivariable logistic regression models were used to estimate adjusted odds ratio (OR) with 95% confidence intervals (CI) for associations between the outcome and socio-demographic variables selected a priori. Kaplan-Meier survival curves were generated for the four categories of HIV-positive patients, depending on relationship between ART and ATT start dates; patients were censored at the end of the data collection period if they were alive and active on ART, or at their last documented visit if they died, were transferred out, or became LTFU either for ATT or ART at any time before the end of the data collection period.

The effect of the introduction of Xpert on mortality outcomes was assessed using a segmented regression analysis of interrupted time series.[19] Deaths observed among TB patients were collected biannually before Xpert (January 2006 to September 2012) and after Xpert (October 2012 to December 2017).

## Ethical considerations

Both the Mozambican National Bioethics Committee for Health (*Comité Nacional de Bioética para a Saúde*, 25/CNBS/2019) and the Institutional Review Board of Vanderbilt University Medical Center (IRB# 190523) approved this analysis. Analysis was performed on de-identified, aggregated patient level data, and no individual informed consent was obtained. The need for written informed consent was explicitly waived from the participants.

## Results

### Patient characteristics

A total of 9,916 patients ≥15 years of age were enrolled at CHC during the study period. Exclusions included (n = 261, 2.6%): 236 transferred out before treatment completion, 18 were duplicate records, and 7 had no ART start date recorded. The complete set of demographic and clinical data was available for the remaining 9,655 patients. Of these, 4,262 (44.1%) were female and median age was 36 years (IQR, 29–46). The majority (89.2%) were new TB cases, with the most frequent type of TB classified as clinically diagnosed PTB (52.0%), followed by bacteriologically confirmed PTB (30.1%) and EPTB (17.9%). HIV testing was conducted in 9,645 patients (99.9%), of which 7,994 (82.8%) were TB/HIV co-infected (Table 1). TB/HIV co-infection was highest in those with EPTB (86.9%; p<0.001). Trends in demographic and clinical characteristics are summarized in Table 2, showing no major variations by year.

### Treatment outcomes

Among all study subjects, 7,068 (73.2%) had a favorable outcome (Table 3), being highest in patients with bacteriologically confirmed TB (74.8%; p<0.01). Female patients (76.1%) had

**Table 1. Sociodemographic and clinical characteristics of TB patients ≥15 years old enrolled in the National TB Control Program at Carmelo Hospital between January 1, 2006 and December 31, 2017.**

| N = 9,655 | All | (%) |
|---|---|---|
| **Gender** | | |
| Female | 4,262 | 44.1 |
| Male | 5,393 | 55.9 |
| **Age** | | |
| 15–24 | 1,036 | 10.7 |
| 25–64 | 8,178 | 84.7 |
| 65+ | 441 | 4.6 |
| **TB Diagnosis** | | |
| New Diagnosis | 8,611 | 89.2 |
| All Others | 1,044 | 10.8 |
| **TB type** | | |
| Clinically diagnosed | 5,018 | 52.0 |
| Bacteriologically confirmed | 2,907 | 30.1 |
| Extrapulmonary | 1,730 | 17.9 |
| **HIV Status** | | |
| HIV (-) | 1,650 | 17.1 |
| HIV (+) | 7,994 | 82.8 |
| Inconclusive/Not done | 11 | 0.1 |
| **HIV/ART status with ATT timing** | | |
| HIV (-) | 1,650 | 17.1 |
| HIV (+) / ART Naïve | 1,186 | 12.3 |
| HIV (+) / ATT before ART | 4,694 | 48.6 |
| HIV (+) / on ART / ATT ≤90 days | 651 | 6.7 |
| HIV (+) / on ART / ATT >90 days | 1,474 | 15.3 |
| **Median distance from health facility in 10 Kms (IQR)** | 0.1 | |

ART–Antiretroviral treatment, ATT–Anti-tuberculosis Treatment

higher rates of favorable treatment outcomes than males (70.9%). Looking at individual outcomes, males had higher mortality (20.7% vs. 16.6%) and LTFU (4.6% vs. 3.4%) (Table 3).

In multivariable logistic regression models, characteristics independently associated with a favorable TB treatment outcome were female sex, younger age, being diagnosed as a new TB case, having bacteriologically confirmed PTB, and being HIV-negative. Males had reduced odds of a favorable TB treatment outcome compared to females (aOR:0.74, 95% CI: 0.67–0.82). Every subsequent year of enrolment into TB treatment was associated with a 3% (95% CI: 1.02–1.05) increased odds of a favorable treatment outcome over the course of the study period (Table 4). Looking at the ART status of patients on TB treatment, it appears that the worst outcomes were for those who never initiated ART, and also for patients that were on ART before treatment start, particularly in the group that was on ART for less than 90 days (Fig 1). Finally, for every 10km increase in distance the patient lived from the health facility, patients had 5% lower odds of a favorable outcome.

## Effect of Xpert on mortality outcomes

Between January 2006 and June 2012, the average number of deaths among TB patients at CHC was 82.8 (SD 16.17) per half year before Xpert was introduced. This decreased to a half year average of 67.4 (SD 21.97) deaths among TB patients in the period between July 2012 and

**Table 2. Characteristics of TB patients by year, at Carmelo Hospital between January 1, 2006 and December 31, 2017.**

| Year | Gender [Males (%)] | Median Age years (IQR) | TB Type [PTB (%)] | HIV status [Positive (%)] | Distance from clinic Km, (IQR) | TB Diagnosis [New (%)] | On ART [n (%)] | Deaths [n (%)] | Total |
|---|---|---|---|---|---|---|---|---|---|
| **2006** | 320 (52.7) | 35 (28–44) | 497 (81.8) | 496 (81.7) | 0.1 (0.1–1.2) | 570 (93.9) | 351 (70.7) | 134 (22.0) | 607 |
| **2007** | 416 (56.3) | 37 (29–47) | 597 (80.8) | 629 (85.2) | 0.5 (0.1–2.0) | 667 (90.3) | 521 (82.8) | 158 (21.4) | 738 |
| **2008** | 378 (51.8) | 34 (28–44) | 618 (84.7) | 594 (81.4) | 0.4 (0.1–2.0) | 664 (91.0) | 472 (79.4) | 158 (21.6) | 729 |
| **2009** | 403 (56.4) | 35 (29–45) | 578 (80.9) | 622 (87.1) | 0.5 (0.1–2.2) | 650 (91.0) | 515 (82.8) | 166 (23.2) | 714 |
| **2010** | 428 (56.3) | 35 (29–45) | 608 (80.0) | 642 (84.4) | 0.1 (0.1–2.0) | 707 (93.0) | 547 (85.2) | 173 (22.7) | 760 |
| **2011** | 579 (54.2) | 35 (29–46) | 924 (86.6) | 885 (82.9) | 0.5 (0.1–2.3) | 964 (90.3) | 744 (84.0) | 201 (18.8) | 1,067 |
| **2012** | 537 (57.5) | 35 (29–46) | 800 (85.7) | 757 (81.4) | 0.1 (0.1–2.2) | 842 (90.2) | 658 (86.9) | 154 (16.5) | 933 |
| **2013** | 418 (54.9) | 34 (29–45) | 594 (78.0) | 630 (82.7) | 0.1 (0.1–2.2) | 691 (90.8) | 533 (84.6) | 86 (11.3) | 761 |
| **2014** | 534 (58.1) | 37 (30–47) | 660 (71.8) | 725 (78.9) | 0.1 (0.1–2.3) | 821 (89.3) | 653 (90.0) | 120 (13.0) | 919 |
| **2015** | 477 (54.5) | 36 (30–46) | 668 (76.3) | 750 (85.7) | 0.1 (0.1–2.0) | 774 (88.5) | 680 (90.4) | 136 (15.5) | 875 |
| **2016** | 482 (59.0) | 37 (31–47) | 636 (77.8) | 679 (83.1) | 0.6 (0.1–2.7) | 665 (81.4) | 628 (92.2) | 149 (18.2) | 817 |
| **2017** | 420 (57.6) | 37 (30–48) | 553 (75.8) | 579 (79.4) | 0.1 (0.1–1.2) | 589 (80.8) | 518 (89.4) | 184 (25.2) | 729 |

IQR–Interquartile range; PTB—Pulmonary TB; ART–Anti-retroviral treatment

December 2017, representing a significant decline in mortality after the introduction of Xpert testing. To assess for chance and control for HIV status and TB type the segmented regression model output shows that deaths among TB patients at CHC decreased on average by 47 patients ($p = 0.01$) after the introduction of Xpert, a result that was statistically significant (Fig 2).

**Table 3. Frequency of favorable treatment outcome, death and lost to follow-up (LTFU) disaggregated by sex and type of tuberculosis, at Carmelo Hospital between January 1, 2006 and December 31, 2017.**

| TB type | Outcome | Total | M | F |
|---|---|---|---|---|
| Extra-pulmonary (n = 1730) | Favorable | 73.9% | 72.7% | 75.2% |
| | Death | 20.0% | 21.2% | 18.5% |
| | LTFU | 2.8% | 2.8% | 2.8% |
| Clinically diagnosed (n = 5018) | Favorable | 72.0% | 68.6% | 76.2% |
| | Death | 20.8% | 23.8% | 17.1% |
| | LTFU | 4.2% | 4.7% | 3.6% |
| Bacteriologically confirmed (n = 2907) | Favorable | 74.8% | 73.6% | 76.4% |
| | Death | 14.9% | 15.1% | 14.6% |
| | LTFU | 4.6% | 5.4% | 3.5% |
| Total (n = 9655) | Favorable | 73.2% | 70.9% | 76.1% |
| | Death | 18.9% | 20.7% | 16.6% |
| | LTFU | 4.1% | 4.6% | 3.4% |

M–male; F–female; LTFU–lost to follow up

**Table 4. Logistic regression model: Predictors of a favorable TB treatment outcome (defined as cured or treatment completed), at Carmelo Hospital between January 1, 2006 and December 31, 2017.**

| Variables | Estimates from Logistic regression models | | | | | |
|---|---|---|---|---|---|---|
| | Unadjusted | | | Adjusted | | |
| | OR (95% CI) | | p-value | aOR (95% CI) | | p-value |
| **Gender** | | | | | | <0.001 |
| Female | 1 | | | 1 | | |
| Male | 0.76 | (0.70–0.84) | <0.001 | 0.74 | (0.67–0.82) | |
| **Age** | | | | | | 0.05 |
| 15–24 | 1 | | | 1 | | |
| 25–64 | 0.78 | (0.67–0.91) | <0.01 | 0.89 | (0.76–1.05) | |
| 65+ | 0.84 | (0.65–1.09) | 0.2 | 0.72 | (0.54–0.94) | |
| **TB Diagnosis** | | | | | | 0.001 |
| New Diagnosis | 1 | | | 1 | | |
| All Others | 0.73 | (0.63–0.83) | <0.001 | 0.76 | (0.65–0.89) | |
| **TB type** | | | | | | 0.02 |
| Clinically diagnosed | 1 | | | | | |
| Bacteriologically confirmed | 1.16 | (1.05–1.29) | <0.01 | 1.16 | (1.03–1.29) | |
| Extrapulmonary | 1.11 | (0.99–1.25) | 0.07 | 1.09 | (0.97–1.25) | |
| **HIV status***[*] | | | | | | |
| HIV (-) | 1 | | | | | |
| HIV (+) | 0.54 | (0.47–0.61) | <0.001 | | | |
| **HIV/ART status with ATT timing**[1] | | | | | | <0.001 |
| HIV (-) | 1 | | | 1 | | |
| HIV (+) / ART never started | 0.14 | (0.12–0.18) | <0.001 | 0.15 | (0.12–0.17) | |
| HIV (+) / on ART at treatment start ≤90 days | 0.37 | (0.30–0.45) | <0.001 | 0.37 | (0.29–0.45) | |
| HIV (+) / on ART at treatment start >90 days | 0.50 | (0.43–0.59) | <0.001 | 0.50 | (0.41–0.59) | |
| HIV (+) / ART Naïve at treatment start but started on ART | 0.90 | (0.78–1.05) | 0.19 | 0.89 | (0.76–1.04) | |
| **Year** | | | | | | <0.001 |
| 2005–2018 (every yearly increase) | 1.03 | (1.01–1.04) | <0.001 | 1.03 | (1.02–1.05) | |
| **Distance from health facility** | | | | | | <0.001 |
| Every 10km increase | 0.95 | (0.93–0.97) | <0.001 | 0.96 | (0.94–0.98) | |

[1]ART–Antiretroviral treatment, ATT–Anti-tuberculosis Treatment

[*]Not included in the model as correlated to HIV status and ART/ATT treatment timing

## Survival estimates

Patients starting ATT before starting ART had a higher probability of survival, while patients not starting ART had the lowest values (Fig 2). Patients starting ATT ≥90 days after ART had a better estimate of survival than patients starting ATT <90 days after ART start, however both groups had lower estimates of survival than patients starting ATT before starting ART.

## Discussion

Favorable TB treatment outcomes were observed for approximately 73% of patients in our cohort, below the 90% reported by the NTP for Gaza Province as a whole in 2018.[12] While our reported LTFU rates were similar to those reported by the NTP, the proportion of patients with documented death was significantly higher at CHC (~19%) compared to the NTP (7%). This difference may be attributed to the fact that CHC is a reference hospital for TB and HIV

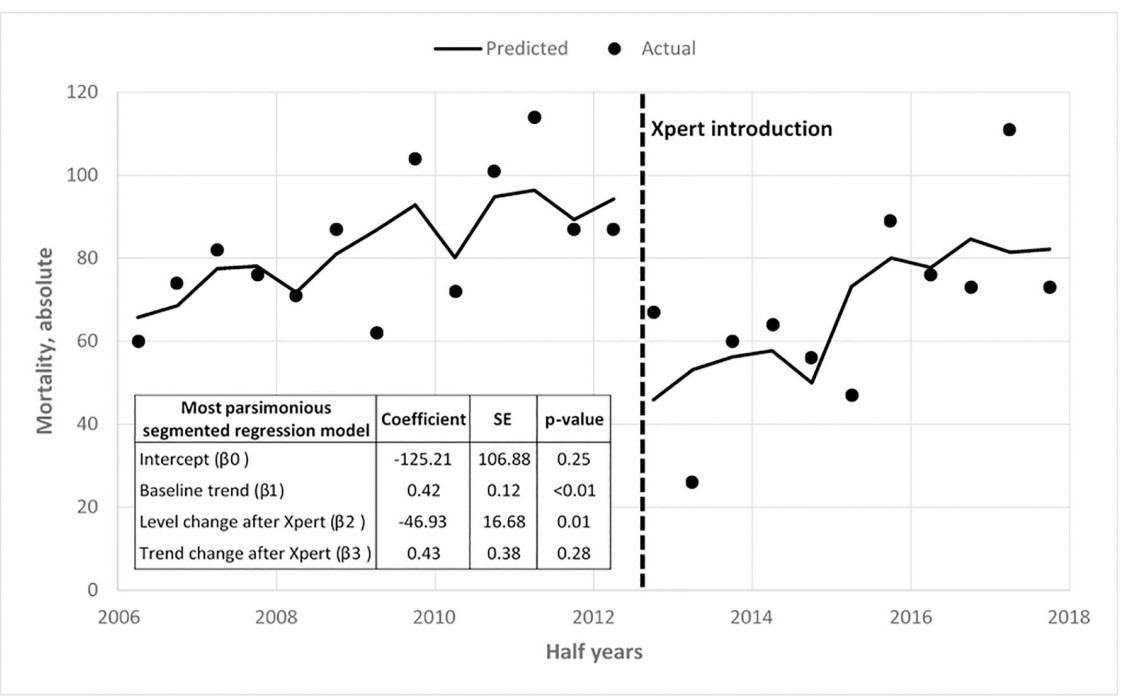

**Fig 1. TB-related mortality at CHC.** Interrupted Time series adjusting for TB type and HIV status–Xpert intervention starts 2012 (July-December).

care for Chókwè and its adjacent districts, with most peripheral health facilities referring complicated TB and HIV cases to CHC.

We found nearly all (99.9%) TB patients enrolled at CHC knew their HIV status, either having tested prior to enrollment or subsequently upon diagnosis of TB. This represents a very positive accomplishment in CHC's contribution to the first 90 of the UNAIDS 90-90-90 goals, superseding the WHO target of 85%, as well as the proportion of TB patients with known HIV status (86%) reported for Africa more generally in 2017.[2,20] CHC has consistently shown strong compliance with national HIV testing guidelines, resulting from focused attention to the provision of quality HIV counseling as well as coordinated efforts for communication between the TB and HIV services. Targeted interventions to sustain high levels of HIV testing at CHC have proven to be very important for the delivery of quality care, as the rates of TB/HIV co-infection are extremely high at 83%, more than double the 36% reported for Mozambique as a whole in 2018.[12] We speculate that the main reasons for this difference is due, again, to CHC´s designation as a reference hospital for both TB and HIV, and to the fact that the population of Gaza Province and specifically Chókwè district engages in cross border migration into South Africa to work in the mines, more than other geographic regions of the country.

Factors that were positively associated with a favorable TB treatment outcome included female gender, having bacteriologically confirmed PTB, being HIV-negative [21,22], and a more recent year of treatment initiation. The difference in treatment outcomes between female and male patients has previously been described[23] but in this case is likely to be compounded by immigration to work in South Africa's mining sector. Male miners come home for treatment when they get sick, but as soon as they feel better, they return to their work, very often without requesting a transfer or communicating the travel to the health staff.

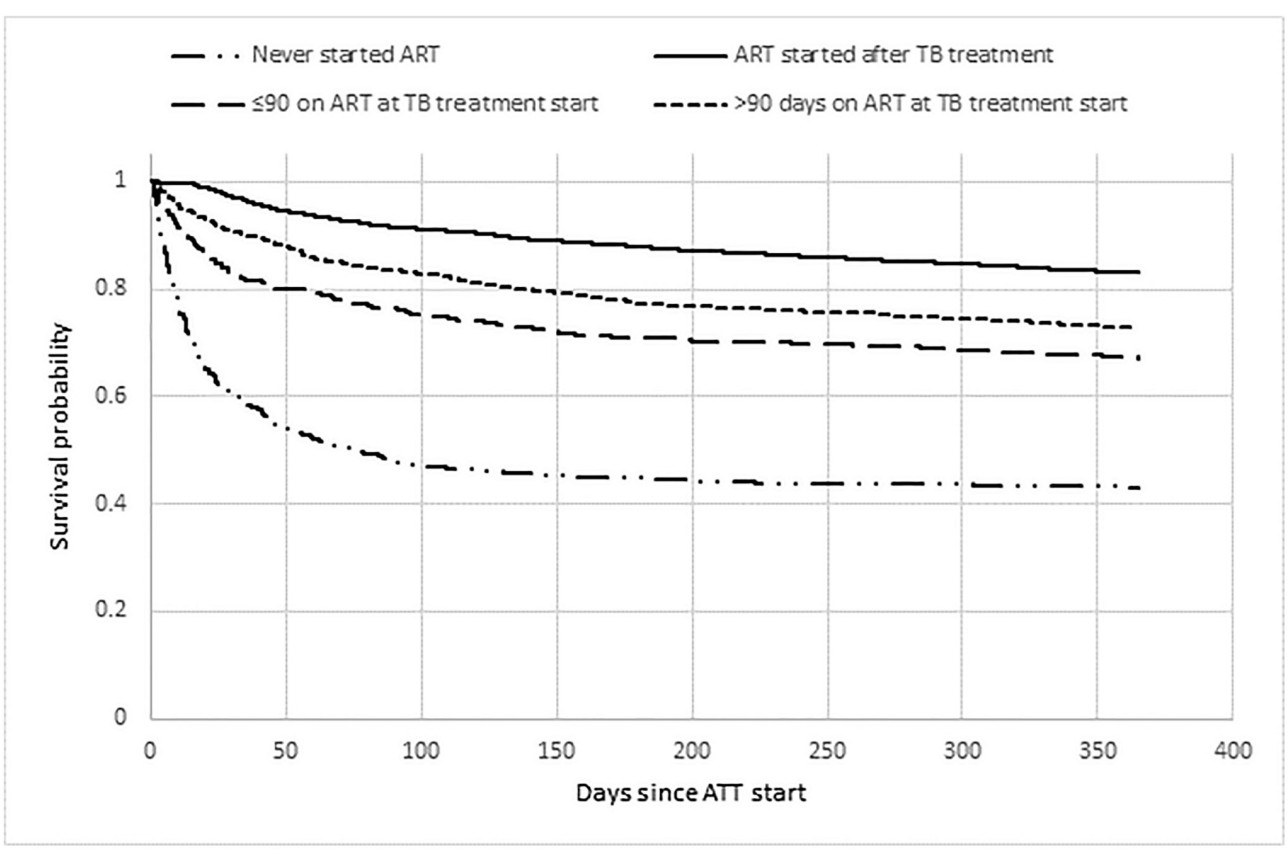

**Fig 2. Kaplan- Meier plot for TB/HIV co-infected patients enrolled at Carmelo Hospital of Chókwè (2006–2017) by ART treatment status in relation to TB treatment.**

Furthermore, better outcomes in more recent years are most likely due to the improvements seen in the delivery of supportive services to persons with TB as these health programs evolved over time, for example improvements in patient follow up by community workers.[24]

Distance from home to clinic has been reported by different authors as a factor influencing adherence and loss to follow up in TB patients, and hence worsening treatment outcomes. [25,26] Transport cost and physical inability to walk long distances to the hospital to pick up the treatment are probably the most influential reasons leading to treatment default.

Favorable treatment outcomes in our cohort increased in the initial 2-year period following introduction of Xpert MTB/RIF, along with a parallel reduction in mortality. This suggests that Xpert-mediated improved and earlier identification of TB patients is a likely contributor to the increase in favorable outcomes, as seen in a recent metanalysis.[27] Even so, other changes to the programme may have also contributed to this improvement of outcomes.

The poor treatment outcomes in those not on ART and those who were on ART, but for less than 3 months, highlights the need for integrated TB and HIV services. We believe that this may be due to the intensity of the introduction of the national HIV "*test and treat*" approach.[28] As this national policy became enacted and programs such as the President´s Emergency Plan For AIDS Relief (PEPFAR) and the Global Fund to Fight AIDS, Tuberculosis, and Malaria began to heavily emphasize its implementation, all health facilities across the country began to see the HIV care and treatment programs flooded with new patients being HIV tested; large numbers of new patients needing initiation of ART; and a concentrated

refocusing of the attention of health workers into implementation of the new strategies, possibly to the detriment of other activities such as screening for TB.

When the "test and treat" approach is not matched by careful clinical TB screening before initiating ART, the result is likely a significant number of active TB cases going unnoticed, mainly in patients with advanced HIV infection. These missed TB cases can appear as IRIS in the first months following ART initiation. IRIS has had clinical presentations which are complicated to manage in resource-limited settings, leading to worse treatment outcomes.[29,30]

The strengths of this study include its large sample size, long study period, and that data were retrieved from detailed registers. The study is extremely powerful in that it documents treatment outcomes in a large routine population, rather than a clinical trial or study setting. This study has the limitation of many retrospective studies where the collection of information is often inadequate and incomplete. It also lacks follow-up information for patients beyond TB treatment completion, particularly in HIV-negative patients. This may have a bearing on reported outcomes especially for death and LTFU. This was not unique to this study, as other studies in Mozambique have reported similar challenges with data completeness.[14,15]

## Conclusion

Introduction of Xpert testing have had a beneficial effect on both favorable treatment outcomes and mortality. This paper highlights the need for TB and HIV care integration with evidence of poor outcomes in those who start TB treatment shortly after ART initiation and those who are never started on ART. There is an appearance of a worsening of treatment outcomes after the start of the "test and treat" approach. A cohort analyses spanning beyond 2017 would be needed to confirm this hypothesis.

## Supporting information

**S1 Table. TB patients Carmelo 2006–2017_anonymized.**
(ZIP)

## Acknowledgments

The authors thank the patients and staff of Carmelo Hospital of Chókwè, Gaza Province, Mozambique, for their co-operation. The authors also want to express their gratitude to the Daughters of Charity of Saint Vincent of Paul for granting the researchers access to hospital facilities and patient records.

## Author Contributions

**Conceptualization:** Edy Nacarapa, Troy D. Moon, Emilio J. Valverde.

**Data curation:** Emilio J. Valverde.

**Formal analysis:** Edy Nacarapa, Evans Muchiri, Troy D. Moon, Salome Charalambous, Emilio J. Valverde.

**Methodology:** Edy Nacarapa, Evans Muchiri, Troy D. Moon, Emilio J. Valverde.

**Resources:** Emilio J. Valverde.

**Supervision:** Emilio J. Valverde.

**Validation:** Edy Nacarapa, Evans Muchiri, Troy D. Moon, Salome Charalambous, Maria E. Verdu.

**Writing – original draft:** Emilio J. Valverde.

**Writing – review & editing:** Edy Nacarapa, Evans Muchiri, Troy D. Moon, Salome Charalambous, Maria E. Verdu, Jose M. Ramos, Emilio J. Valverde.

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
