## [Decision Letter · Decision Letter 0]

24 Jan 2020

PONE-D-19-35601

Tuberculosis treatment among HIV adults on ART in rural southern Mozambique. A retrospective cohort study.

PLOS ONE

Dear Dr. Valverde,

Thank you for submitting your manuscript to PLOS ONE. After careful consideration, we feel that it has merit but does not fully meet PLOS ONE’s publication criteria as it currently stands. Therefore, we invite you to submit a revised version of the manuscript that addresses the points raised during the review process.

We would appreciate receiving your revised manuscript by Mar 09 2020 11:59PM. To enhance the reproducibility of your results, we recommend that if applicable you deposit your laboratory protocols in protocols.io, where a protocol can be assigned its own identifier (DOI) such that it can be cited independently in the future. For instructions see: http://journals.plos.org/plosone/s/submission-guidelines#loc-laboratory-protocols

We look forward to receiving your revised manuscript.

Kind regards,

HASNAIN SEYED EHTESHAM

Academic Editor

PLOS ONE

Additional Editor Comments (if provided):

Minor Revision

Reviewers' comments:

Reviewer's Responses to Questions

**Comments to the Author**

1. Is the manuscript technically sound, and do the data support the conclusions?

Reviewer #1: Yes

Reviewer #2: Yes

2. Has the statistical analysis been performed appropriately and rigorously? 

Reviewer #1: I Don't Know

Reviewer #2: Yes

3. Have the authors made all data underlying the findings in their manuscript fully available?

Reviewer #1: Yes

Reviewer #2: Yes

4. Is the manuscript presented in an intelligible fashion and written in standard English?

Reviewer #1: Yes

Reviewer #2: Yes

5. Review Comments to the Author

Reviewer #1: The manuscript by Nacarapa et. al, describes the use of Xpert MTB/RIF technology in tuberculosis diagnosis in a resource-limited, high-burden TB/HIV coinfection settings. The authors used a large sample size over a long time for this study. This manuscript addresses an important and timely topic regarding clinical TB screening before initiating ART in TB/HIV co-infection cases. This additional information can be useful in designing more effective treatment strategy for this particular risk group.

The authors have described some positive factors associated with a favorable TB treatment. It would be important to know, why and how these factors are linked to favorable TB treatment. I think the authors should critically discuss such aspects along with suitable literature references to support their views. This would also tremendously increase the value of the manuscript.

Reviewer #2: 1. The title of the paper does not justify the aim of the study and should be modified accordingly.

2. The authors should include information about rate of mortality in TB patients and TB- HIV co-infected patients before and after introduction of genexpert (Year-wise data would be better).

3. The authors should add few lines explaining the scheme of treatment followed in TB and TB-HIV co-infected patient in their country. The authors can also add information about if any changes has been incorporated and why the changes were done and in which year they were done.

4.Does any government health agency/NGO monitors the treatment and diagnosis of TB ? A Brief information about the policy for controlling TB in the country will provide better idea to the readers.

6. PLOS authors have the option to publish the peer review history of their article (what does this mean?). If published, this will include your full peer review and any attached files.

Reviewer #1: No

Reviewer #2: No

---

## [Author Response · Author response to Decision Letter 0]

17 Feb 2020

Dear reviewers,

Thank you very much for your very informative comments. We have attached a point by point response to your comments along with the manuscript.

---

## [Editor Report · Decision Letter 1]

20 Feb 2020

Effect of Xpert MTB/RIF testing introduction and favorable outcome predictors for tuberculosis treatment among HIV infected adults in rural southern Mozambique. A retrospective cohort study.

PONE-D-19-35601R1

Dear Dr. Valverde,

We are pleased to inform you that your manuscript has been judged scientifically suitable for publication and will be formally accepted for publication once it complies with all outstanding technical requirements.

With kind regards,

HASNAIN SEYED EHTESHAM

Academic Editor

PLOS ONE

Additional Editor Comments (optional):

I have gone through this revised manuscript and also the Author response to the comments of the Reviewers. The authors have satisfactorily addressed all the comments made by the reviewers and added more information, treatment regimens and have revised the manuscript accordingly.

I recommend this manuscript for publication.
---

## [Editor Report · Acceptance letter]

28 Feb 2020

PONE-D-19-35601R1 

Effect of Xpert MTB/RIF testing introduction and favorable outcome predictors for tuberculosis treatment among HIV infected adults in rural southern Mozambique. A retrospective cohort study. 

Dear Dr. Valverde:

I am pleased to inform you that your manuscript has been deemed suitable for publication in PLOS ONE. Congratulations! Your manuscript is now with our production department. 

With kind regards,

on behalf of

Prof HASNAIN SEYED EHTESHAM 

Academic Editor

PLOS ONE